# Validation and tuning of *in situ* transcriptomics image processing workflows with crowdsourced annotations

**Jenny M. Vo-Phamhi, Kevin A. Yamauchi**⓭**, Rafael Gómez-Sjöberg**⓭\*

Chan Zuckerberg Biohub, San Francisco, California, United States of America

\* rafael.gomez@czbiohub.org

## Abstract

Recent advancements in *in situ* methods, such as multiplexed *in situ* RNA hybridization and *in situ* RNA sequencing, have deepened our understanding of the way biological processes are spatially organized in tissues. Automated image processing and spot-calling algorithms for analyzing *in situ* transcriptomics images have many parameters which need to be tuned for optimal detection. Having ground truth datasets (images where there is very high confidence on the accuracy of the detected spots) is essential for evaluating these algorithms and tuning their parameters. We present a first-in-kind open-source toolkit and framework for *in situ* transcriptomics image analysis that incorporates crowdsourced annotations, alongside expert annotations, as a source of ground truth for the analysis of *in situ* transcriptomics images. The kit includes tools for preparing images for crowdsourcing annotation to optimize crowdsourced workers' ability to annotate these images reliably, performing quality control (QC) on worker annotations, extracting candidate parameters for spot-calling algorithms from sample images, tuning parameters for spot-calling algorithms, and evaluating spot-calling algorithms and worker performance. These tools are wrapped in a modular pipeline with a flexible structure that allows users to take advantage of crowdsourced annotations from any source of their choice. We tested the pipeline using real and synthetic *in situ* transcriptomics images and annotations from the Amazon Mechanical Turk system obtained via Quanti.us. Using real images from *in situ* experiments and simulated images produced by one of the tools in the kit, we studied worker sensitivity to spot characteristics and established rules for annotation QC. We explored and demonstrated the use of ground truth generated in this way for validating spot-calling algorithms and tuning their parameters, and confirmed that consensus crowdsourced annotations are a viable substitute for expert-generated ground truth for these purposes.

**Data Availability Statement:** The iPython notebooks, input images, and In Situ Transcriptomics Annotation (INSTA) pipeline software are available from https://github.com/czbiohub/instapipeline. The SpotImage software is

## Author summary

To understand important biological processes such as development, wound healing, and disease, it is necessary to study where different genes are expressed in a tissue. RNA molecules can be visualized within tissues by using in situ transcriptomics tools, which use

available from https://github.com/czbiohub/spotimage. The files and iPython notebooks used to generate the figures for this paper are available from https://github.com/czbiohub/instapaper.

**Funding:** The authors have declared that no competing interests exist.

**Competing interests:** The authors have declared that no competing interests exist.

fluorescent probes that bind to specific RNA target molecules and appear in microscopy images as bright spots. Algorithms can be used to find the locations of these spots, but ground truth datasets (images with spots located to high accuracy) are needed to evaluate these algorithms and tune their parameters. While the typical way of generating ground truth datasets is having an expert annotate the spots by hand, many in situ transcriptomics image datasets are too large for this. However, it is often easy for non-experts to identify the spots with minimal training.

In this paper, we present an open-source toolkit and framework that incorporates crowdsourced annotations alongside expert annotations as a source of ground truth for the analysis of in situ transcriptomics images. We explored and demonstrated the use of this framework for validating spot-calling algorithms and tuning their parameters, and confirmed that consensus crowdsourced annotations are a viable substitute for expert-generated ground truth.

This is a *PLOS Computational Biology* Software paper.

## Introduction

Diversity of form follows diversity of function in biological tissues. The anatomy and cellular properties of each tissue come from cell-specific gene expression patterns.[1] To understand important biological processes, such as development, wound healing, and disease, it is necessary to study the 3-dimensional spatial architecture of biological tissues and their gene expression patterns at the cellular (or even subcellular) level. Recent advancements in *in situ* methods[2–9] (e.g., DNA[10–13], RNA[10,14,15], and protein[10,16] measurements in tissue sections) have deepened our understanding of the way biological processes are spatially organized in tissues. In particular, recent *in situ* transcriptomics tools, such as multiplexed *in situ* RNA hybridization and *in situ* RNA sequencing, have facilitated the spatial mapping of gene expression with subcellular resolution.[1]

*In situ* transcriptomics methods utilize the binding of fluorescent probes to specific RNA target molecules with high complementarity within cultured cells and tissue sections. Extracting the positions of the fluorescent probes, which appear in microscopy images as bright spots, presents a key image processing challenge. Automated spot detection is not trivial due to noise arising from light scattering and background autofluorescence, plus spot crowding.[17] Although automated image processing and spot-calling algorithms exist (for brevity, from now on, we will use the term "spot-calling algorithm" to refer to the whole image processing and spot-finding pipeline), they have many parameters which need to be tuned for optimal detection.[18–22] Having ground truth datasets (images where there is very high confidence on the accuracy of the detected spots) is essential for evaluating these algorithms and tuning their parameters.

Studies typically use synthetic images to evaluate or test the performance of any spot detector because ground truth does not inherently exist for real images.[17] The typical way of generating ground truth datasets for real images is having an expert inspect the images and annotate the valid spot locations by hand.[17] In cases where manual annotation of a large *in situ* transcriptomics image dataset by an expert is unfeasible, it is necessary to have alternative sources of ground truth. One proposed solution to this problem is iterative human-in-the-loop

deep learning workflows, where ground truth generated by spot-calling algorithms can be manually refined.[23] Since valid spot locations can often be apparent even to minimally-trained, non-expert human eyes, we propose that crowdsourcing is a feasible way to generate large, high quality ground truth datasets.

Crowdsourcing refers to the use of web-based systems to recruit random volunteers or paid workers to perform tasks remotely. Recent work has indicated that carefully crowdsourced annotations can expedite data processing tasks that have a visual component. Volunteer-based citizen science has made substantial contributions to areas of biology from proteomics[24–26] to ecology.[27] When tasks are less intrinsically interesting to volunteers, minimally-trained workers can complete tasks for small payments through crowdsourcing platforms such as Amazon's Mechanical Turk (MTurk), and the consensus annotations (across multiple workers or "turkers") can be highly comparable with expert annotations, and sufficiently reliable for use as training data for detection algorithms.[27–29] Therefore, we hypothesized that consensus from crowdsourced annotations can be used as a substitute for ground truth to tune and benchmark spot-calling algorithms. However, there are no published *in situ* transcriptomics pipelines that can incorporate ground truth from crowdsourced annotations. Such pipelines should have mechanisms to prepare images for annotation, process annotations, establish consensus from the annotations, and generate annotation performance metrics.

In this paper, we present INSTA (*IN situ* Sequencing and Transcriptomics Annotation), an open-source toolkit and framework that incorporates crowdsourced annotations alongside expert annotations as a source of ground truth for the analysis of *in situ* transcriptomics images. Using real images from *in situ* experiments, and simulated images produced by a tool we developed to generate synthetic, customizable *in situ* transcriptomics images, we explored worker sensitivity to the size, quantity, and density of spots, and we established rules for annotation quality control. Based on these rules, we developed tools for preparing images to optimize workers' ability to annotate these images reliably, performing quality control (QC) on worker annotations to get maximum value from them, extracting candidate parameters for spot-calling algorithms from sample images, tuning parameters for spot-calling algorithms, and evaluating spot-calling algorithms and worker performance. We wrapped these tools in a modular pipeline with a flexible structure that allows users to take advantage of crowdsourced annotations. The toolkit includes an annotation ingestion class designed to work with Quanti. us[28], and it can be easily adapted to work with any crowdsourcing system by creating custom annotation ingestion classes. We tested the pipeline using images from the *in situ* transcriptomics dataset from starfish[30,31], a Python library for analysis of image-based transcriptomics data developed by the Chan Zuckerberg Initiative, and annotations from MTurk via Quanti.us. We explored and demonstrated the use of ground truth generated in this way for validating spot-calling algorithms and tuning their parameters, and confirmed that consensus crowdsourced annotations are a viable substitute for expert-generated ground truth for these purposes.

A user would incorporate INSTA into a wider in-situ sequencing transcriptomics workflow as follows (S1 Fig): (1) Run test experiment with the chosen chemistry. (2) Check SNR and spot density (in the Results section, we provide guidance on the characteristics which make an image suitable for INSTA). (3) Expert annotates some images. (4) Use these annotations to try tuning a spot-finding algorithm. (5) Decide if crowd-sourcing annotations are necessary. (6) If so, consider running more experiments to get more images. (7) Use INSTA to prepare the images for crowdsourcing the annotations. (8) Send out images for annotations. (9) Receive annotations and process them with INSTA. (10) Tune and benchmark a spot-finding algorithm using the consensus annotations.

In addition to this pipeline, we created a tool to generate synthetic *in situ* images, which we call SpotImage. This tool receives background images from real *in situ* experiments and adds simulated spots to them. The user can vary spot characteristics including size, shape, location, distribution throughout the image, and signal to noise ratio (SNR). These parametrized synthetic images are very useful for testing crowdsourced annotations and spot calling algorithms. More details in S1 Text.

## Materials and methods

### Structure of a modular pipeline for tuning *in situ* transcriptomics image processing with crowdsourced annotations

This section provides a high-level overview of INSTA (Fig 1). Greater detail will be provided in Sections III and IV, and examples with two different *in situ* transcriptomics chemistries are provided in Sections V and VI.

The input to the pipeline consists of images from a particular *in situ* transcriptomics chemistry and a spot detection algorithm to be optimized for that chemistry. An expert designates one image as a representative image for the *in situ* chemistry used, and annotates it. The remaining images are assigned to the test dataset.

From this small amount of expert input, the tool learns approximately what a spot in this chemistry should look like. That is, a script extracts parameters which characterize the brightness (intensity) and size (sigma of a 2D Gaussian approximation of a spot) profiles of the spots of that chemistry. These parameters are passed to a spot detector that uses scikit-image's implementation of the Laplacian of Gaussian algorithm.[18]

The pipeline then processes each test image separately. For each test image:

○. The spot detector uses the spot parameters it learned from parameter extraction to do rough, first-pass spot-calling.

○. A script detects the crowded regions and recursively crops the images until the sub-images are sufficiently uncrowded that a human worker should easily be able to click on all the spots without getting frustrated or tired.

○. All the pieces of the image–the crops and the parent images–are sent to Quanti.us, which is a platform for crowdsourced image annotation through Amazon's Mechanical Turk platform. Custom crowdsourced data ingestion classes can be written to allow the pipeline to work with any crowdsourced annotation system.

○. Each image is annotated by a user-defined number of workers (typically 20 to 30). In the annotation analysis stage, all the crowdsourced annotations are clustered and QC is performed based on characteristics of the spots and clusters to produce consensus annotations, which are then reassembled to produce an original image that has been annotated with high precision and recall.

○. These annotations can be used to tune and train other spot detectors. They can also be used to validate or quantify the performance of the spot detector that this run of the pipeline attempted to optimize, or of another detector.

○. If the optimized spot detector's performance is satisfactory, the detector may be useful for other images. If the detector's performance is unsatisfactory, the parameters can be modified and the detector can be reevaluated against the worker annotations.

Two key aspects of the pipeline should be highlighted: First, individual segments of the pipeline may be used separately for assorted purposes. For example, to simply get annotations

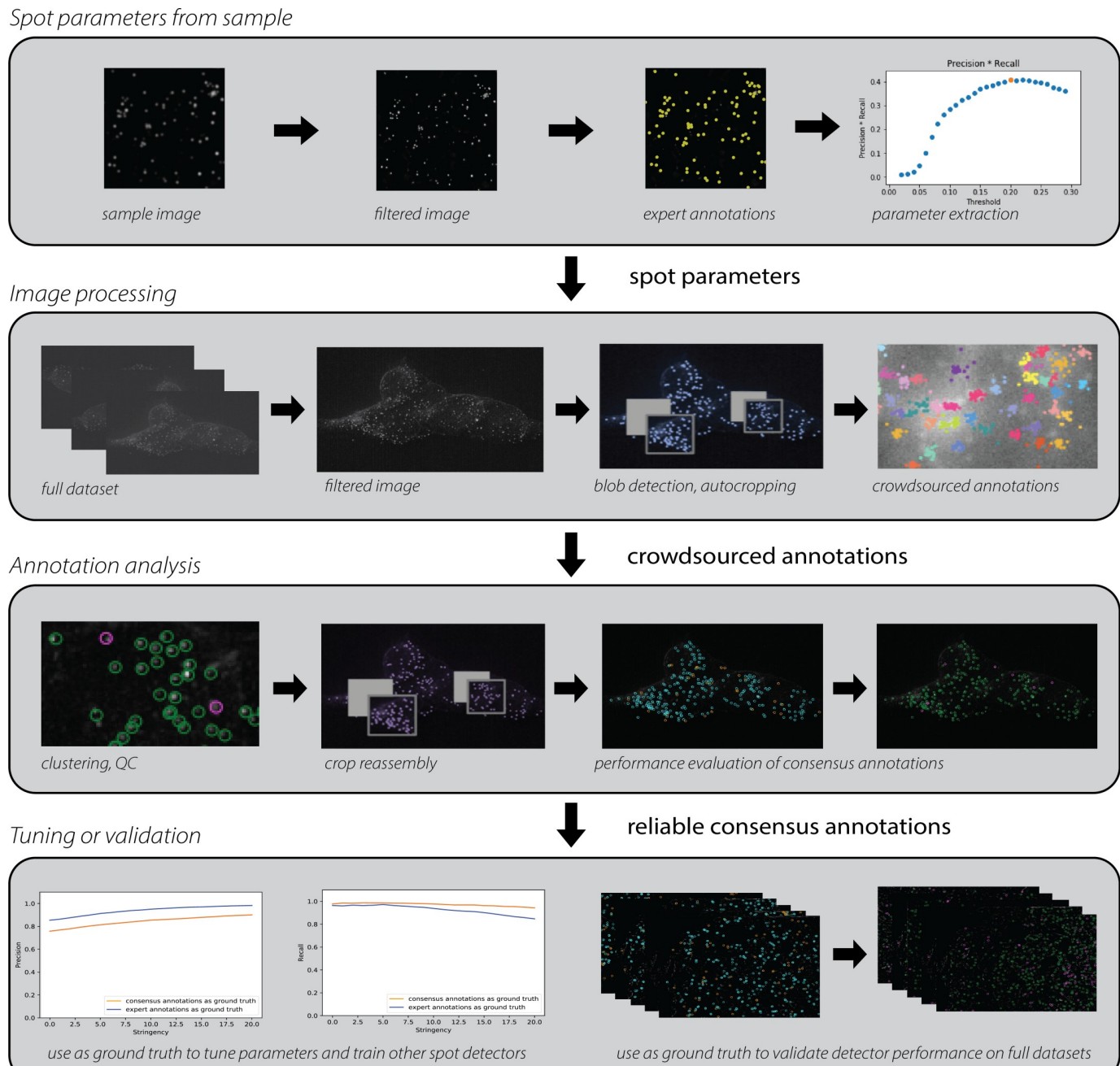

**Fig 1. INSTA (IN situ Sequencing and Transcriptomics Annotation) is a pipeline for tuning and validating spot detection methods using crowdsourced annotations.** The steps for optimizing the spot detection parameters specific to a given chemistry are as follows. Row 1: A sample image from that chemistry is filtered and annotated by an expert. From this small amount of expert input, the tool learns what a spot in this chemistry should look like and passes these parameters to a spot detector. Row 2: The images that will receive consensus annotations are individually pre-processed. The spot detector uses the learned parameters to do rough, first-pass spot-calling. A function then detects the crowded regions and recursively zooms in to them until the sub-images are sufficiently uncrowded that a human worker should easily be able to click on all the spots without getting fatigued. Then, all the sub-images and parent images are sent to Quanti.us, which is a system for crowdsourced annotation through Amazon's Mechanical Turk platform. Crowdsourced workers annotate each crop. Row 3: Quality control is performed based on characteristics of the spots and clusters to produce consensus annotations, which are then reassembled to produce an original image that has been annotated with high precision and recall. Row 4: These annotations can be used to tune and train other spot detectors. They can also be used to quantify the performance of the spot detector that this run of the pipeline attempted to optimize.

for images without optimizing any spot detectors, the latter portion of the pipeline can be used to crop images and QC the crowdsourced annotations, with cropping based on spots detected by some detector that has been deemed sufficiently good for the purpose of preliminary detection. Second, workers tend to perform poorly on full-size raw images because there are too many spots and the spots are too close together. The pipeline includes a recursive cropping functionality that automatically breaks up each image into sections that the workers can handle effectively.

Section III will discuss the limits of what workers can accurately annotate with regard to brightness, density, number of spots, etc. Section IV will further discuss ways to prepare images to optimize worker performance.

## Results

### Worker performance is limited by spot crowding, visibility, and quantity

Given the large number of in situ transcriptomics technologies and even larger number of tissues to analyze, we sought to understand the fundamental properties of the images required for high quality crowd-sourced annotations. In particular, we aimed to characterize worker performance as a function of signal-to-noise ratio (SNR) and the distances between spots. We hypothesized that these two properties of in situ transcriptomics images would be fundamental determinants of annotation quality.

Many factors in sample preparation and imaging affect the SNR of spots in in situ transcriptomics images. For example, some tissues exhibit higher autofluorescence owing to the composition of the cells and extracellular matrix.[32,33] Light scattering due to tissue composition further degrades SNR, with shorter wavelengths generally more degraded than longer wavelengths.[34] Understanding how the protocol chemistry and tissue properties affect performance can help guide the sample preparation optimization process. For example, optimizing sample preparation via tissue clearing [35] and fixative selection [36] can improve SNR and contrast.

The ability to resolve adjacent spots depends on both their physical distances and the magnification and microscope quality used to image them. The physical distances between mRNA molecules in the cell vary both by their abundance and localization. Dense transcripts can be imaged with higher magnification, which can improve resolution at the cost of imaging speed. Alternatively, expansion microscopy can be used to improve the spatial resolution of transcripts.[37,38] Anticipating that dense transcripts could present a challenge for workers to annotate, we developed quality control methods to identify and declump annotations from dense spots, and created the automatic image subdivision system.

We crowdsource annotations using Quanti.us (30) and perform clustering and quality control on the annotations to arrive at consensus coordinates for spot locations. Each image sent to Quanti.us is annotated by 25 workers (S2 Text). The resulting annotations are then clustered via Affinity Propagation to find the initial set of annotation clusters—this algorithm does not require *a priori* knowledge of the number of clusters.[39,40] Given that some of the annotations do not correspond to spot locations and some of the annotations cover adjacent spots (S4 Fig), we next perform QC to identify false positives and unmix adjacent clusters.

To determine if a cluster is a false positive, we threshold the clusters by the number of annotations in the cluster (Fig 2A). In annotations of synthetic images we observed that clusters are distributed bimodally (S3 Text) by number of annotations. Clusters with few annotations tend to be incorrect (that is, the cluster centroid is not within a given threshold pixel radius of the closest actual spot location). So for the first QC step, clusters are sorted by number of annotations and one-dimensional $k$-means with $k = 2$ is applied to find the threshold number of

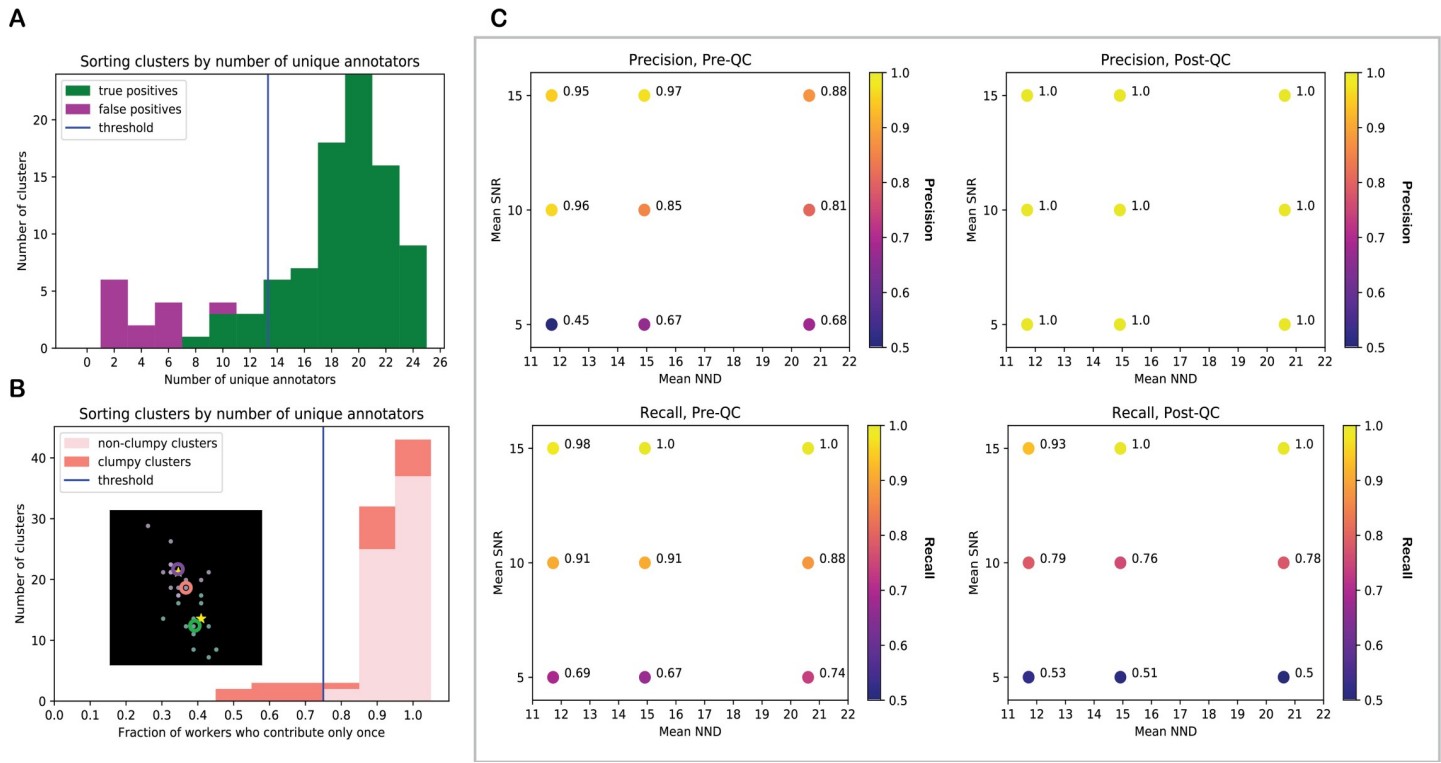

**Fig 2. QC, including cluster size thresholding and declumping, improves precision, sometimes at the expense of recall, for images with lower SNR values. (A)** Clusters with fewer workers tend to be incorrect. Sort clusters by number of unique workers annotating them. The fraction of workers who contribute once can predict whether a cluster is clumpy (it corresponds to multiple image spots that are close together). **(B)** Sort clusters by fraction of unique workers contributing. Isolate and declump the clusters where many workers contribute more than once, as the inset demonstrates using real data. (Inset–orange circle: original centroid, green and purple circles: new centroids found by declumping, green and purple dots: worker annotations assigned to new centroids by declumping, stars: actual spot locations.) Clumps are declumped using 2D k-means. For more examples of declumped clusters, see S2 Fig. **(C)** Thresholding clusters by the number of annotations in the cluster and by the fraction of unique workers who contribute multiple times to the cluster improved recall by 17%, while decreasing precision by 11% on average, in an experiment with nine images with 50, 100, and 150 spots; mean SNR = 5, 10, and 15; and average nearest neighbor distance (NND) ≈ 11.5, 15, and 20.5.

annotations. All clusters with fewer annotations than this threshold are removed. This thresholding method tends to be aggressive; we would rather miss spots than "detect" incorrect spots, since in reality it is inevitable that some spots will be missed anyway when signals are too faint or overlap. In an experiment with nine synthetic images (with 50, 100, and 150 spots and mean SNR of 5, 10, and 15), this QC step yielded 100%, 100%, and 100% precision (40%, 13%, and 7% increase compared with the precision obtained without thresholding clusters by number of annotations) and 51%, 78%, and 99% recall (19%, 12%, and 0% decrease) for images with mean SNR = 5, 10, and 15 respectively. For each SNR, the reported precision and recall value is the average across the three images with different numbers of spots (S5A Fig).

To detect whether a cluster corresponds to multiple spots that are very close together, we threshold the clusters by the fraction of unique workers who contribute multiple times to the cluster. We observed that when spots are very close together, the clusters associated with multiple spots may clump together into one annotation cluster, but some workers do detect that the spots are supposed to be separate and those workers contribute more than one click within the region that the clustering algorithm detects as one cluster. (See pink clusters in S6 Fig.) The threshold fraction is found between the main mode of the distribution and the tail by identifying the point of steepest increase in histogram values. All clusters with a greater fraction of multiple-clicking workers than this threshold fraction are removed. Therefore, the fraction of

workers who contribute only once can predict whether a cluster is actually clumpy, even if sometimes the actual spots are so close that most of the workers looking at them interpret them as one spot and it is not possible to identify that cluster as clumpy (Fig 2B). We declump each clumpy cluster using two-dimensional $k$-means (Fig 2B). In the same experiment with synthetic images, this QC step yielded 64%, 88%, and 94% precision (4%, 1%, and 0% increase over results without QC) and 67%, 87%, and 96% recall (3%, 3%, and 3% decrease) for images with mean SNR = 5, 10, and 15 respectively (S5B Fig).

Performing declumping after false positive detection boosts recall. In the same experiment with synthetic images, removing small clusters and separating clumpy clusters yielded 100%, 100%, and 100% precision (40%, 13%, and 7% increase in precision) and 51%, 78%, and 98% recall (19%, 12%, and 2% decrease in recall) for images with mean SNR = 5, 10, and 15 respectively (Fig 2C).

We observed some limits on the utility of QC. When spot visibility was poor, nearest neighbor distance was very small, or an image had too many spots, the quality of the raw annotations was very low, so QC was unable to improve precision and recall to acceptable levels.

Using the NND-varying and SNR-varying features of our SpotImage tool, we tested the limits of spot visibility and crowdedness that workers can accurately annotate. In an experiment with synthetic images of mean SNR = 3, 5, 7, 9, and 11, and spot size = 0.5, 1.0, and 1.75 pixels (S7A Fig), small spot sizes required larger mean SNR to achieve recall greater than 50%. For example, when the spot size (sigma of a gaussian fit to the intensity) was half a pixel (about 5 microns), recall was zero until SNR > 9 (S7B Fig). At lower SNR values, even spots with large nearest neighbor distances tended to be missed, and as spot SNR increased the mean NND of undetected spots decreased (S7C Fig). While the size of cellular structures in pixels will be a function of the imaging system and chemistry used, note that the cells in the smFISH image used for Fig 3C are 250 to 300 pixels wide, and 130 to 180 pixels tall. A typical nucleus from an RCA image (e.g. idr0071-feldman-crisprko/experimentE/10X_A11_Tile-0.aligned from Feldman et al. 2018 [41]) is approximately 8–12 pixels in diameter.

We observed that the radius of the symbol that Quanti.us uses to mark spots identified by workers limits the minimum nearest neighbor distance between spots which we can reasonably expect workers to discern to 4% of the image's width. When spots are closer than this distance, the mark on one spot obscures neighboring spots. We tested inverting the images (dark spots on a light background) before submitting them to Quanti.us to see if this would improve worker performance. A linear regression between recall with inversion and recall without inversion resulted in a slope of 1.004, with a Pearson's correlation coefficient of $r$ = 0.985 (S7D Fig), making it clear that inversion is not helpful.

We also sought to understand the total number of spots that workers can annotate in one image. In an experiment with SNR = 3, 5, 7 and number of spots per image between 50 and 225, the number of clicks per worker per image increased as the number of spots in the image increased, until it leveled off at around 120 on average, suggesting that 120 was the upper bound on the total number of spots workers were willing to click for the payment offered in these experiments ($0.05 per image) (S8A Fig). For example, in an assay detecting 31 genes in fresh-frozen breast cancer tissue, Ke et al. report an average of 25 spots per cell [42], which means that a worker can reliably annotate about 4.8 cells per image. If the average number of spots per cell for one gene is instead 16 (Fig 2B in [43,44]), the same worker would be able to annotate 7.9 cells per image. As the number of spots increased beyond 50, the fraction of spots that workers were willing to click decreased. On average, workers annotated almost all spots for images with 50 spots but only about 60% of all spots for images with 200 spots (S8B Fig). However, even though each worker annotated a smaller fraction of the spots as the number of

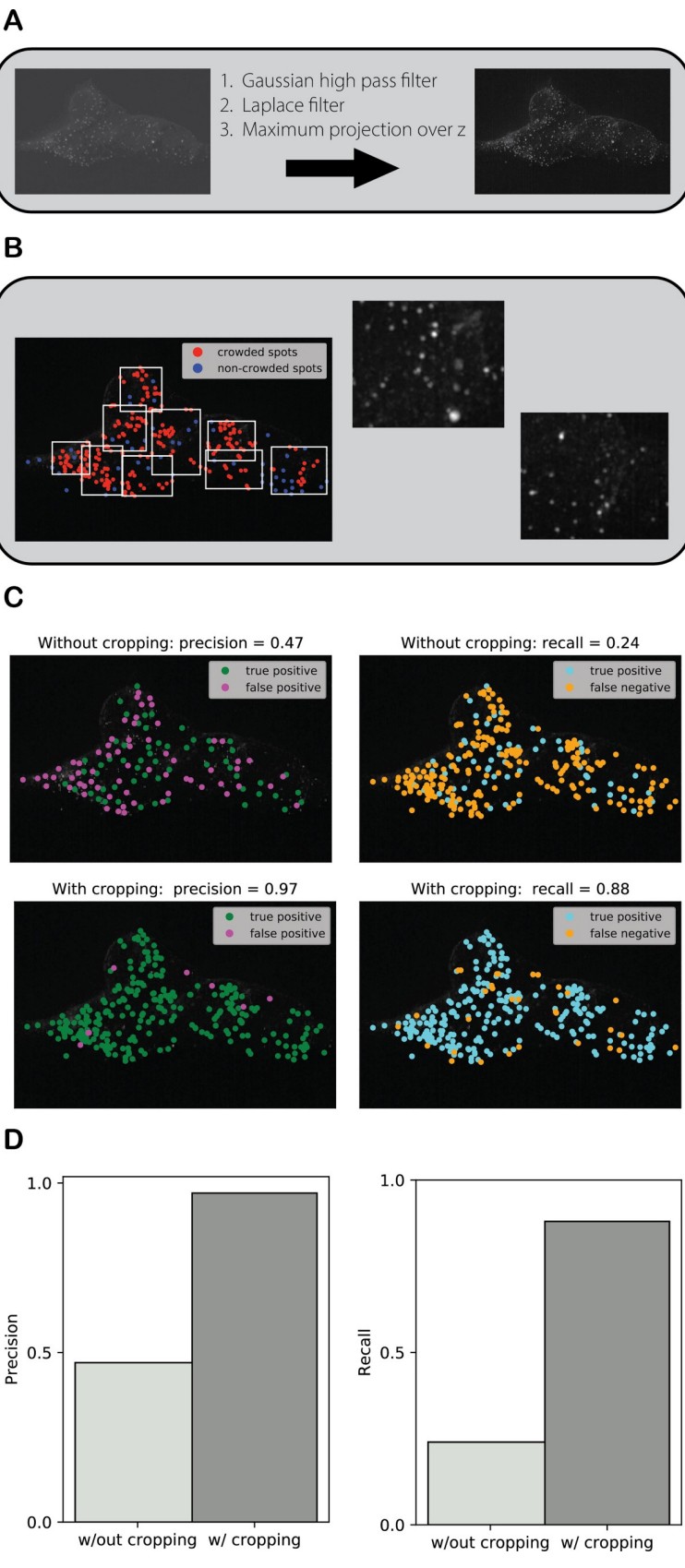

**Fig 3. Images are filtered and subdivided so that they are easier for workers to annotate. (A)** The raw image (which contains 268 spots) is pre-processed with a gaussian high pass filter, a Laplace filter, and a maximum intensity projection over z. **(B)** Crowded spots detected and bound. Rough, first-pass spot-calling enables the detection of crowded spots and subsequent automatic subdivision into smaller images. **(C)** True positive = consensus in concordance with expert annotation, false positive = consensus not in concordance with expert annotation, and false negative = no consensus found for an expert annotation. The distance between a correct consensus annotation and the nearest expert annotation is no more than the user-defined correctness threshold. The distance between a detected expert annotation and the nearest consensus annotation is also no more than the user-defined correctness threshold. **(D)** Applying cropping resulted in precision and recall of 97% and 87%, improvements of 50% and 64%, respectively, over the un-cropped image.

spots in the image increased, most spots were still getting annotations from at least half the workers (S8C Fig). In other words, some workers detected spots which other workers missed.

These effects of spot visibility, crowding, and quantity on worker performance are intuitive, but these results provide quantitative metrics to understand worker performance as a function of image quality, and objective guidelines for image pre-processing that allow us to optimize the performance for a given image dataset. We found that for best performance the spot signal-to-noise ratio must be at least 10 and the mean nearest neighbor center-to-center spot distance must be between 12 and 15 pixels. The performance characterization in this section provides targets for the SNR and density of spots that can aid in guiding the optimization of *in situ* technologies to new samples. In the next section, we present the methods we developed to prepare images which workers are more likely to annotate reliably.

## Image pre-processing improves workers' ability to annotate images reliably

Given the guidelines we identified for the spot density, total number of spots and number of spots that can be accurately annotated in a given image, raw *in situ* transcriptomics images would need to be pre-processed in order to meet the guidelines.

In the first preparation step, the raw images are pre-processed with filters to enhance contrast between spots and the rest of the image (Fig 3A). This process is described in greater detail in S4 Text. In the second step, the filtered images are automatically subdivided to produce child images with sufficiently small spot densities and large nearest neighbor distances to be effectively annotated by workers (Fig 3B). S4 Text provides more details about preprocessing. In an experiment with a real single molecule fluorescence *in situ* hybridization (smFISH) image with 268 spots of typical contrast and density, annotated by 25 workers, applying cropping resulted in precision and recall of 97% and 87%, improvements of 50% and 64% respectively, over the un-cropped image (Fig 3C).

There are limits on how much an image can be cropped to increase recall. Some spots are so close together that the spots themselves appear to clump in the parent image, so no amount of zooming in would enable them to be separated. The user should also keep in mind that in the Quanti.us website cropped images are upscaled by interpolation to a size which workers can annotate, so the pixels in the upscaled image which the workers are annotating are not a perfect representation of the original data. The user should also consider the relationship between the number of crops and the overall cost of the experiment. Annotations from Quanti.us cost only five cents per worker per image (in 2020), but many workers may be needed to ensure good coverage and proper clustering and de-clumping of clusters.

## Helper images

Because images from experiments using RCA (Rolling Circle Amplification, an *in situ* transcriptomics chemistry which is described in S5 Text) have a lot of variation in spot sizes, we sought to investigate whether we could improve the quality of the worker annotations by

providing better guidance at the crowdsourcing interface, and whether including the parent image with the crops removed in the stack (e.g. second image from left, S9B Fig) affects worker accuracy. The parent image appears to the workers on the Quanti.us interface at the same size as the crops, so the spots in the parent image appear much smaller to the workers than the spots in the crops. Additionally, we designed four different helper images–two variants, with or without circles drawn around the correct spots–which illustrate what the workers should click on (S11 Fig).

For RCA image ISS_rnd1_ch1_z0, which contains 287 spots, the inclusion of helper images on average increased precision by 14% (95% with helper images and 81% without) and decreased recall by 16% (59% with helper images and 76% without) (S12 Fig). When the spots in the helper image were circled, precision was 0.4% higher and recall was 3.4% higher. Workers expressed little preference between the two variants of helper images. On average, precision and recall with images of the first variant were only 0.5% greater and 4% less than precision and recall of the second variant, respectively. However, including the parent image in the stack, which workers would also annotate, decreased both precision and recall by 6% and 4% respectively.

These results suggest that the workers were less likely to click spots they felt unsure about when helper images were provided. These results also demonstrate that for some images it is disadvantageous to show the workers the parent image, as it confuses them because of the drastic difference in scale between the parent image and the crops.

These image preparation steps complete the pipeline described in Section II (Fig 1). The performance of the pipeline will now be demonstrated with a vignette, using the rolling circle amplification (RCA) chemistry.

## The results of validating spot calling algorithms using crowdsourced ground truth are comparable to the results using expert annotations

We tested the usage of crowdsourced annotations resulting from the pipeline as ground truth to validate spot calling algorithms. Using available images from RCA experiments, we sought to evaluate how well crowdsourced annotations agree with expert annotations to assess the generalizability of the tuned spot-calling parameters.

To produce the crowdsourced annotations, the inputs to the annotation generation pipeline were: One sample image with the RCA chemistry, expert spot location annotations for that image, and three test images without annotations. All images were downloaded from an *in situ* sequencing (ISS) experiment in the starfish database.[45] The spots which an expert had annotated in the sample image were analyzed to extract spot detection parameters required by starfish's BlobDetector method, which implements the Laplacian of Gaussian spot detection approach. These parameters were used for first-pass spot detection on the test images, and the resulting spot coordinates were used to automatically subdivide the test images. All crops were then sent to Quanti.us to be annotated by 25 workers each. Further details are explained in S6 Text.

We sought to evaluate how well the resulting crowdsourced annotations agreed with expert annotations. Precision and recall for the consensus annotations, based on an expert's evaluations of the test images, were 95% and 70%, 92% and 89%, and 81% and 76% for images ISS_rnd0_ch1_z0, ISS_rnd0_ch3_z0, and ISS_rnd1_ch1_z0, which contain 1236, 416, and 287 spots respectively (Fig 4). The Jaccard similarity indices (intersection over union) between the consensus annotations and the expert annotations were 71%, 85%, and 68%, respectively. We then sought to understand how well the consensus annotations and the expert annotations agreed with each other by evaluating the level of concurrence between different experts and

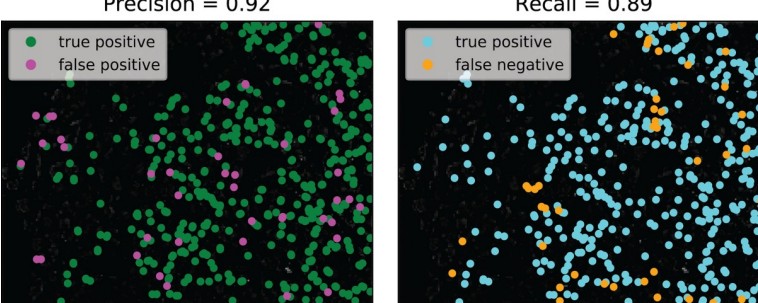

**Fig 4. The in situ transcriptomics annotation pipeline was tested using RCA (Rolling Circle Amplification) images from an in situ sequencing (ISS) experiment in the starfish database.** Worker consensus annotations for RCA test image ISS_rnd0_ch3_z0, which contains 416 spots, achieved 92% precision and 89% recall based on expert consensus annotations. The Jaccard similarity index (intersection over union) between the consensus annotations and the expert annotations was 0.85.

between the same expert annotating the same image twice six months apart (at times $t_0$ and $t_1$ = $t_0$ + 6 mos) (S7 Text). The Jaccard indices (intersection over union) for Expert #1 at $t_0$ and Expert #1 at $t_1$, Expert #1 at $t_0$ and Expert #2 at $t_1$, and Expert #1 at $t_1$ and Expert #2 at $t_1$ were 73%, 78%, and 82% respectively (S13 Fig). Thus, the agreement we saw between the consensus and expert annotations for the RCA images are in the same range as the intra- and inter-expert Jaccard indices. We also observed that consensus annotations are more likely to include false positives for specks of debris that experts would ignore (S14A Fig). These specks are smaller and fainter than the spots an expert would annotate. However, this common error mode in the consensus annotations does not necessarily hamper the performance of a spot detector trained on consensus annotations. For image ISS_rnd0_ch1_z0, the blobs found using scikit-learn's blob_log() algorithm with parameters based on worker consensus annotations actually had higher precision and recall (84.8% and 73.1%) than the blobs found using scikit-learn's blob_log() algorithm with parameters based on expert annotations (75.1% and 72.8%) (S14B Fig). While recalls differ by only 0.4%, precision is significantly lower (12.1% difference) for spots found based on expert consensus because the expert identified spots with a larger range of sizes and intensities than the workers. Consequently, parameter tuning that took these edge cases into consideration caused blob_log() to capture more false positives and more spots overall (2019 vs. 1388).

We also sought to assess the generalizability of the tuned spot-calling parameters and the potential utility of using crowd-sourced annotations to generate ground truth for a large number of images. To do this, we ran starfish's BlobDetector method using the spot parameters which had been extracted in this vignette on thirteen other images from starfish's RCA dataset which had not been annotated by experts. By visual inspection, these images had a greater variation in spot size, brightness, and quantity than the other images. As ground truth, we used consensus annotations for these images. The precision of the BlobDetector method was 82 ± 10%, and the recall was 68 ± 17% (mean ± standard deviation, S16 Fig). These results suggest that when a set of spot parameters tuned to a particular channel and field of view for a chemistry are used for other channels and fields of view for the same chemistry, the spots detected are likely to be correct but fewer spots may be detected.

The finding that the Jaccard indices for the consensus and expert annotations for the RCA images were similar to the intra- and inter-expert Jaccard indices strongly suggests that consensus annotations can be used in place of expert annotations. We also found that spot parameters found for a given chemistry using consensus annotations as ground truth can be used to

automatically find spots with precision (82 ± 10%, mean ± standard deviation) comparable to the agreement between two experts annotating the same image (82%). Together, these findings are especially useful for the processing of large datasets that would be infeasible for an expert to annotate by hand.

### Crowdsourced ground truth is useful for tuning and validating spot calling parameters

This section tests the usage of crowdsourced annotations resulting from the pipeline as ground truth to tune and validate their parameters. Ground truth is essential for testing how well a spot-calling algorithm generalizes to other *in situ* transcriptomics chemistries and tuning parameters (S8 Text). We tested whether consensus and expert annotations function similarly well as ground truth to tune parameters for spot-calling algorithms, using the RCA dataset from starfish.[31] We also bootstrapped our consensus and expert annotations for the RCA dataset to explore the minimum number of ground truth annotations needed to effectively tune a spot-calling algorithm.

Usage of consensus annotations and expert annotations as ground truth to tune the LMP (starfish's LocalMaxPeakfinder) stringency parameter, which controls the intensity cutoff to detect a peak, resulted in similar precision vs. stringency trends, as well as recall vs. stringency trends (Fig 5). The optimal stringency parameter found using consensus annotations from crowdsourced workers alone as ground truth resulted in a lower precision and slightly higher recall (89.4% and 95.4% respectively, compared with 94.3% and 94.8% from using expert annotations alone as ground truth– 5.3% and 0.63% difference respectively) for RCA image ISS_rnd0_ch1_z0, which contains 1236 spots. These results reflect the fact that some of the spots annotated by the worker consensus were not annotated by the more conservative expert.

The optimal stringency parameter found using consensus annotations from crowdsourced worker annotations pooled together with the expert's annotations resulted in precision and recall almost equivalent to the results using consensus annotations from crowdsourced workers alone (89.5% and 95.3%– 0.1% percent difference for both). This result likely reflects the fact that the number of expert annotations is typically much smaller than that of worker annotations, so their effect on the pool is minimal. Nevertheless, combining crowdsourced annotations with expert annotations provides a way to extend the amount of ground truth available.

S9 Text shows that the training behavior of the expert and crowdsourced annotations is very similar. That is, performance converges to roughly the same level and at roughly the same rate. When blob detection was executed on RCA image ISS_rnd0_ch1_z0, which contains 1236 spots, fifteen ground truth annotations (spots) were enough to get 99.1% and 98.1% of the maximum precision performance when the annotations were produced by experts and worker consensus, respectively. With the same number of annotations, 97.6% and 96.6% of the maximum recall performance was achieved with annotations produced by experts and worker consensus, respectively. Together with the result that consensus annotations and expert annotations resulted in similar precision vs. stringency and recall vs. stringency trends when used as ground truth to tune the LMP stringency parameter (Fig 5), this suggests that consensus annotations are a viable substitute for expert-generated ground truth for both parameter tuning and algorithm validation.

## Discussion

We developed INSTA, a pipeline to prepare *in situ* transcriptomics images for crowdsourced annotation and integrated these pipeline components into an open-source toolkit which takes an *in situ* image dataset or a spot detector as input, prepares the images for crowdsourcing

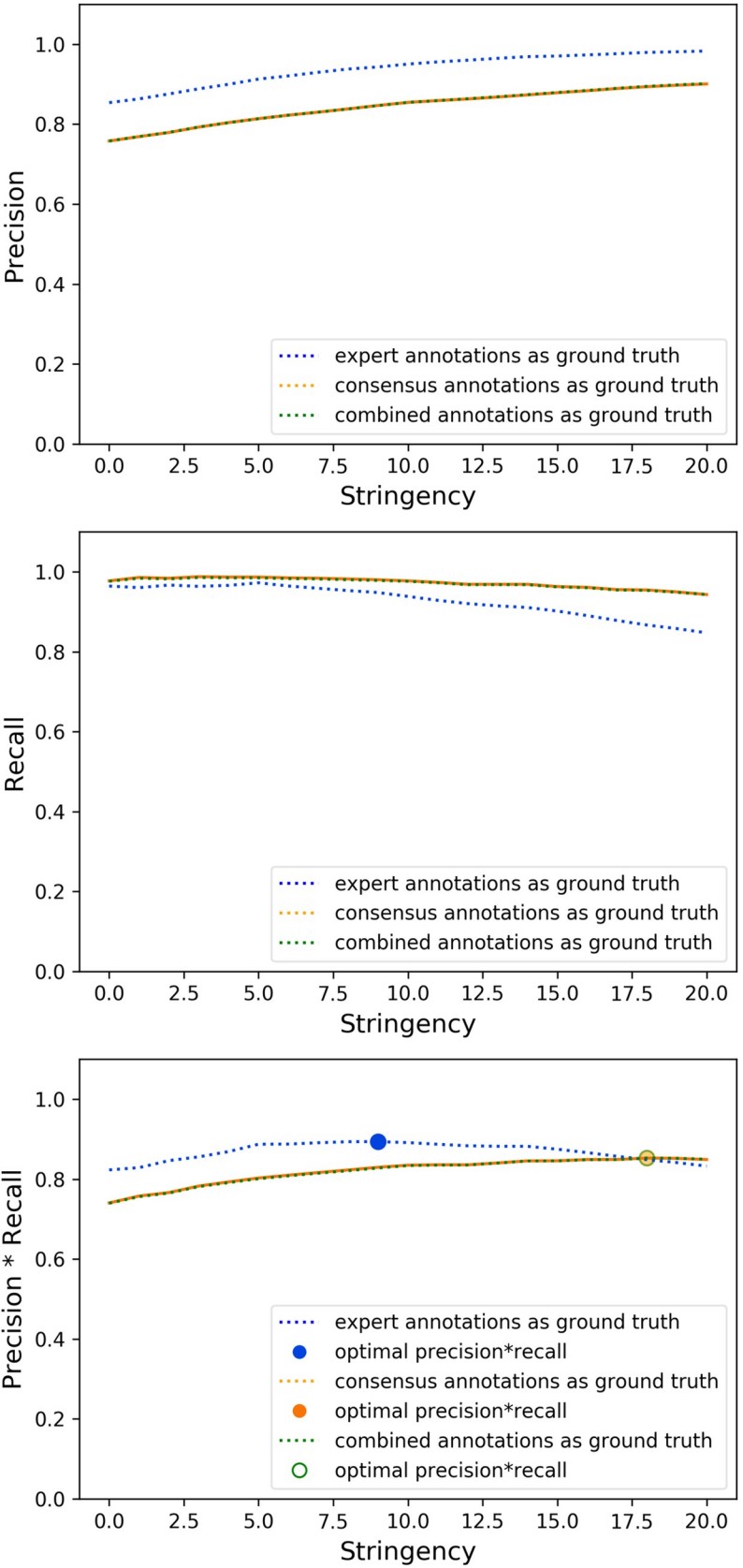

**Fig 5. The optimal "stringency" parameter for starfish's LocalMaxPeakfinder (LMP) with Rolling Circle Amplification (RCA) images from the starfish database resulted in lower precision and slightly higher recall (89.4% and 95.4% respectively) for RCA image ISS_rnd0_ch1_z0, which contains 1236 spots, when consensus annotations from crowdsourced workers alone were used as ground truth for parameter tuning, compared with 94.3% and 94.8% precision and recall which were achieved when expert annotations alone were used as ground truth for parameter tuning.** The optimal stringency parameter found using consensus annotations from crowdsourced worker annotations pooled together with the expert's annotations resulted in precision and recall almost equivalent to the results using consensus annotations from crowdsourced workers alone (89.5% and 95.3%– 0.1% percent difference for both).

annotation, receives the annotations, and outputs consensus locations for the spots and/or optimized parameters for the detector. The pipeline was designed to be flexible enough to easily include components of the user's choice (e.g. custom crowdsourced annotation ingestion classes) and to accommodate manual user intervention at different points in the pipeline. We also created a tool (SpotImage) to generate simulated *in situ* images with adjustable parameters, such as spot density and SNR. Using simulated and real *in situ* transcriptomics images, we developed QC rules for crowdsourced annotations based on observable aspects of the annotation data, such as cluster characteristics, and used these rules to develop QC methods to optimize consensus precision and recall. We also gained insight into critical aspects of how the quantity, size, SNR, and crowdedness of spots in images all influence worker behavior and thus annotation quality.

We demonstrated the pipeline using RCA images, resulting in consensus annotations with high precision and recall compared to expert annotations. We also demonstrated that consensus and expert annotations are equally suitable as ground truth. While consensus annotations are useful when large amounts of ground truth are needed to check or validate the performance of spot-calling algorithms, a few dozen expert annotations alone may be sufficient to tune a spot-calling algorithm such as Starfish's BlobDetector, even if that would not be enough to properly validate the algorithm (obtain statistically significant measures of precision and recall). Crowdsourced ground truth is vital for validating algorithm performance on a large dataset with many images, or even just one image with thousands of spots.

The pipeline only requires user intervention for the reviewing and editing of automatically subdivided images, but we believe it is particularly important for the user to have the option to intervene at this stage rather than leave the pipeline as an end-to-end black box. While it would be theoretically ideal to automatically learn cropping parameters which maximize accuracy, minimize the number of crops used (and therefore cost incurred), and generalize perfectly to all images of a given chemistry, spot distribution characteristics vary too much between the individual *in situ* transcriptomics images of most chemistries for this to be reasonable. The researcher needs the opportunity to balance the tradeoff between crop detail and crowdsourcing cost. If a cheap experiment yields a very large dataset with many images, a user may be less concerned with maximizing data extracted from each image, but if each image costs more to produce, the researcher might wish to be more detailed with cropping (S10 Text).

Even without budgetary constraints, subdividing is only useful to a certain extent because zooming and scaling crops up to the size where they can be displayed for annotation inherently involves interpolation; at a certain point the workers may be annotating a crop that is not faithful to the original image. The toolkit user should be able to intervene before this point.

INSTA can be used to annotate publicly-available image datasets, especially for researchers who do not have access to wet labs. The sample data available through starfish would be a good starting point. We will update pipeline usage and guidelines if we experiment with

processing images of other, more challenging chemistries, and strive to make the pipeline usage as generalizable to other chemistries as possible.

## Supporting information

**S1 Fig. Incorporation of INSTA into a wider in-situ sequencing transcriptomics workflow.**
(TIF)

**S2 Fig. Cluster declumping improves precision and recall by splitting clusters that correspond to multiple actual spots.** Shown here with synthetic spot images generated by the Spot-Image tool on mouse lung image background. (Image background omitted here for clarity of figure).
(TIF)

**S3 Fig. At least 20 annotators are needed per image.** Using simulated spot images, we found that at least 20 workers are necessary to consistently yield precision and recall greater than 95% for images which contain 75 spots, and that the number of workers required for reliable annotation of an image does not increase dramatically as the number of spots in the image increases. All spot images used for this analysis were simulated with spots of SNR = 10 over mouse lung tissue background images. Each marker value represents the average across 10 groups of workers.
(TIF)

**S4 Fig. Some of the worker annotations from Quanti.us do not correspond with spot locations (examples indicated with red arrows) and some of the annotations cover adjacent spots, so quality control is necessary to identify false positives and unmix adjacent clusters.**
(TIF)

**S5 Fig. Thresholding clusters by number of annotations improves precision at the expense of recall, while thresholding by fraction of unique workers who contribute multiple times to the cluster improves recall at the expense of precision. (A)** Thresholding clusters by the number of annotations in the cluster improved precision by 17% while decreasing recall by 10% on average in an experiment with images of mean SNR = 5, 10, and 15 and average NND = ~ 11, 15, and 19. **(B)** Thresholding clusters by the fraction of unique workers who contribute multiple times to the cluster improved recall by 2% while decreasing precision by 1% on average in an experiment with images of mean SNR = 5, 10, and 15 and average NND = ~ 11.5, 15, and 20.5.
(TIF)

**S6 Fig. When spots are very close together, some workers actually do detect that the spots are supposed to be separate and contribute more than one click to a location that the clustering algorithm detects as one cluster.** Markers are only shown for actual clumps.
(TIF)

**S7 Fig. Workers are sensitive to spot visibility (signal to noise ratio) and crowdedness (nearest neighbor distance). (A)** This experiment used synthetic images with spot SNR ranging from 1 to 11; spot size = 0.5, 1.0, and 1.75; plus inverted and not inverted. A subset of these images are shown here. **(B)** In an experiment using these images, small spot sizes required larger mean SNR to achieve good recall. **(C)** At lower SNR values, even spots with large nearest neighbor distances tended to be missed, and as spot SNR increased, the median NND of undetected spots decreased. **(D)** A linear regression between recall with inversion and recall without inversion resulted in a slope of 1.004 with Pearson's correlation coefficient of r = 0.985. In

other words, there was insufficient evidence that inverting the images improved worker performance.
(TIF)

**S8 Fig. As the number of spots in an image increases, the fraction of workers annotating each spot decreases, but most spots still get annotations from a majority of workers (spot SNR = 10). (A)** The number of clicks per worker per image increased as the number of spots in the image increased until it leveled off around 120 on average, suggesting that 120 was the upper bound on the number of spots workers were willing to click. **(B)** As the number of spots increased, the fraction of spots that workers were willing to click decreased. On average, workers annotated almost all spots for images with 50 spots but only about 60% of all spots for images with 200 spots. **(C)** Even though the workers annotated a smaller fraction of the spots as the number of spots in the image increased, most spots were still getting annotations from at least half the workers.
(TIF)

**S9 Fig. Results from a representative dataset as it progresses through the pipeline. (A)** The inputs to the pipeline were one sample image with the RCA chemistry, expert spot location annotations for that image, and three test images without annotations. **(B)** Blob detection provided a general idea of the regions where spots were located and images were automatically subdivided. **(C)** Annotations were crowdsourced, QC'd, and reassembled as consensus.
(TIF)

**S10 Fig. Parameter extraction yields spot parameters that can be used by spot calling algorithms. (A)** The largest sigma associated with a spot annotated by the "expert" is designated sigma_max, which indicates the maximum spot size of a spot that can be detected. **(B)** The threshold parameter, which indicates the lower bound on the brightness of a detected spot, is chosen which optimizes precision times recall when blob_log() is executed on the sample image.
(TIF)

**S11 Fig. Helper images accompany rolling circle amplification (RCA) images.** Top and bottom rows: design variants 1 and 2, respectively. Left and right columns: with and without circles drawn around correct spots.
(TIF)

**S12 Fig. Inclusion of helper images improves precision at the expense of recall.** On average, the inclusion of helper images For RCA image ISS_rnd1_ch1_z0, which contains 287 spots, the inclusion of helper images on average increased precision by 14% (95% with helper images and 81% without) and decreased recall by 16% (59% with helper images and 76% without). When the spots in the helper image were circled, precision was 0.4% higher and recall was 3.4% higher. Workers expressed little preference between the two variants of helper images. On average, precision and recall with images of the first variant were only 0.5% greater and 4% less than precision and recall of the second variant, respectively. However, including the parent image in the stack decreased both precision and recall by 6% and 4% respectively.
(TIF)

**S13 Fig. Intra- and inter-expert agreement is comparable to agreement between expert and consensus annotations.** To better contextualize the performance of the consensus annotations, we evaluated the level of concurrence we should expect among experts. The original ($t_0$) expert annotations which have been used as ground truth to evaluate the consensus annotations for the three RCA test images were compared with two new sets of annotations produced

for the same data half a year later ($t_1$): one set produced by the same expert (Expert #1) and one set produced by another expert (Expert #2). The Jaccard similarity indices (intersection over union) for Expert #1 at $t_0$ and Expert #1 at $t_1$, Expert #1 at $t_0$ and Expert #2 at $t_1$, and Expert #1 at $t_1$ and Expert #2 at $t_1$ were 73%, 78%, and 82% respectively.
(TIF)

**S14 Fig. Error modes.** (A) Consensus annotations are more likely to include false positives for specks of debris that experts would ignore. (B) However, for an image from the RCA dataset, the blobs found using scikit-learn's blob_log() algorithm with parameters based on worker consensus annotations had higher precision (84.8%) than the blobs found using the same algorithm with parameters based on expert annotations (75.1%).
(TIF)

**S15 Fig. Starfish's BlobDetector algorithm performs worse than the LocalMaxPeakFinder with cyclic-ouroboros single molecule fluorescence in situ hybridization (osmFISH).** While Starfish's BlobDetector algorithm detected RCA spots with high precision and recall when parameters were optimized (as demonstrated in Section V, precision and recall for the consensus annotations were 95% and 70%, 92% and 89%, and 81% and 76% for images ISS_rnd0_ch1_z0, ISS_rnd0_ch3_z0, and ISS_rnd1_ch1_z0 respectively), the same algorithm performed poorly with osmFISH (21), failing to find a threshold parameter which yielded a precision*recall score better than 0.1219 (precision = 20.7%, recall = 59.0%).
(TIF)

**S16 Fig. Spot calling parameters tuned for an image of a particular chemistry can be used for other fields of view with that chemistry.** To assess the generalizability of the tuned spot-calling parameters, we ran Starfish's BlobDetector method using the spot parameters which had been extracted in this vignette on thirteen other images from Starfish's RCA dataset which had not been annotated by experts. As ground truth, we used consensus annotations for these images. The mean precision was 82% with a standard deviation of 10%, and the mean recall was 68% with a standard deviation of 17%. These results suggest that when a set of spot parameters tuned to a particular channel and field of view for a chemistry are used for other channels and fields of view for the same chemistry, the spots detected are likely to be correct but fewer spots may be detected.
(TIF)

**S17 Fig. Training behavior was very similar between the expert and consensus annotations.** Moreover, when blob detection was executed on ISS_rnd0_ch1_z0, an RCA image from the Starfish dataset that contains 1236 spots, only about 15 or 20 ground truth annotations were needed in order to assure sufficient coverage across the range of spot sizes and intensities and thus get reliable spot size parameters, regardless of whether parameters had been extracted from either expert or consensus ground truth annotations. Above 15 or 20 ground truth annotations, using more sample spots did not significantly improve precision and recall percentage, which leveled off in the high eighties and mid nineties respectively.
(TIF)

**S1 Text. SpotImage for generating synthetic in situ transcriptomics images.**
(DOCX)

**S2 Text. Parameters for Quanti.us annotation.**
(DOCX)

**S3 Text. Sensitivity and specificity of cluster size thresholding on synthetic in situ transcriptomics images.**
(DOCX)

**S4 Text. Preprocessing images before annotation.**
(DOCX)

**S5 Text. Rolling circle amplification (RCA).**
(DOCX)

**S6 Text. RCA annotation pipeline test.**
(DOCX)

**S7 Text. Intra- and inter-expert concurrence.**
(DOCX)

**S8 Text. Ground truth is essential for testing how well a spot-calling algorithm generalizes to other *in situ* transcriptomics chemistries and tuning parameters.**
(DOCX)

**S9 Text. The training behavior of the expert and crowdsourced annotations is very similar.**
(DOCX)

**S10 Text. Balancing the tradeoff between crop detail and crowdsourcing cost.**
(DOCX)

## Acknowledgments

We would like to thank all members of the Chan Zuckerberg Biohub and especially the Bioengineering Platform for their useful discussions and feedback. We also thank Quanti.us for their support, the Chan Zuckerberg Initiative starfish team for helpful discussions, and all the crowd workers for their diligent contributions to the development and testing of the INSTA pipeline.

## Author Contributions

**Conceptualization:** Kevin A. Yamauchi, Rafael Gómez-Sjöberg.

**Data curation:** Jenny M. Vo-Phamhi, Kevin A. Yamauchi.

**Formal analysis:** Jenny M. Vo-Phamhi.

**Funding acquisition:** Rafael Gómez-Sjöberg.

**Investigation:** Jenny M. Vo-Phamhi, Kevin A. Yamauchi.

**Methodology:** Kevin A. Yamauchi.

**Project administration:** Rafael Gómez-Sjöberg.

**Resources:** Kevin A. Yamauchi, Rafael Gómez-Sjöberg.

**Software:** Jenny M. Vo-Phamhi, Kevin A. Yamauchi.

**Supervision:** Kevin A. Yamauchi, Rafael Gómez-Sjöberg.

**Validation:** Jenny M. Vo-Phamhi.

**Visualization:** Jenny M. Vo-Phamhi.

**Writing – original draft:** Jenny M. Vo-Phamhi, Kevin A. Yamauchi, Rafael Gómez-Sjöberg.

**Writing – review & editing:** Jenny M. Vo-Phamhi, Kevin A. Yamauchi, Rafael Gómez-Sjöberg.

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
