## [Decision Letter · Decision Letter 0]

10 Jan 2021

Dear Dr. Gomez-Sjoberg,

Thank you very much for submitting your manuscript "Validation and tuning of in situ transcriptomics image processing workflows with crowdsourced annotations" for consideration at PLOS Computational Biology.

As with all papers reviewed by the journal, your manuscript was reviewed by members of the editorial board and by several independent reviewers. In light of the reviews (below this email), we would like to invite the resubmission of a significantly-revised version that takes into account the reviewers' comments.

We cannot make any decision about publication until we have seen the revised manuscript and your response to the reviewers' comments. Your revised manuscript is also likely to be sent to reviewers for further evaluation.

Sincerely,

Manja Marz

Software Editor

PLOS Computational Biology

Manja Marz

Software Editor

PLOS Computational Biology

Reviewer's Responses to Questions

**Comments to the Authors:**

Reviewer #1: The authors describe an annotation pipeline to crowdsource spot calling and compare the results to expert annotations. The workflow employs quanti.us and Amazon’s Mechanical Turk and assesses a number of steps around data presentation and filtering on the annotation results.

The work could be relevant for researchers interested in annotating and analyzing in situ transcriptomics images and publicly-available image datasets, particularly for processing large datasets that would not be feasible to annotate by hand. It also offers ground truth datasets, which are crucial for assessing and tuning automated image processing and spot-calling algorithms designed to process and analyze in situ transcriptomics images.

Overall, the proposed pipeline is rigorously developed, the experiments are well designed, and the manuscript is generally well structured and written. From a technical perspective, the overall approach is sound. However, I have some comments and concerns.

What I miss in the paper is, first and foremost, a discussion on how this falls in a wider in-situ sequencing transcriptomics workflow.

As the authors indicate, results from either crowdsourced or expert annotated images will be used to train and validate spot detectors. As this paper discusses how expert annotation compares to crowdsourced annotation, I would expect - apart from a statistical quantification of how precise crowdsourced annotation is - an investigation of the impact on the training of spot detectors - i.e., - are there any implicit biases between experts and non-experts that impact downstream training of the model? Moreover, given that this can be applied in a wide variety of biological and experimental settings, I would feel exploration of the following questions should also be addressed:

- What is the impact of different types of tissue. For example, do crowdsourced annotations fare equally well in complex, heterogeneous tissue with varying levels of pathology?

- Different genes will have different RNA location patterns throughout a cell. There are RNA species predominantly nuclear vs. cytoplasmic. E.g., different genes will have a different proclivity to clump. Does this have any impact?

- There are many in-situ sequencing technologies available. How well does crowdsourcing apply to the different technologies?

- Do different imaging settings have an impact (e.g., different fluorescence channels)?

Moreover, it would also be relevant that the authors design and provide a framework to integrate both consensus annotations and expert annotations and investigate how such fused annotations could improve the results. For example, at least for the starfish database, they could report the combination of both consensus annotations and expert annotations as grand truth; then, the cross posting results could also be incorporated into the current version of Figure 5.

Few extra comments:

Figure 1 & Figure 5: Please provide better titles and more descriptive legends.

Figure 2b & Suppl. Figure 1, 3, and 5:

Please provide higher resolution and/or separate images for the raw signal & the annotations. It is not easy to interpret. In Suppl. Figure 1, the colors are difficult to distinguish.

Suppl. Figure 6:

Do the authors not mean images with a spot SNR running from 1 to 11?

Reviewer #2: ‘Validation and tuning on in situ transcriptomic image processing workflows with crowdsourced annotations’ by Vo-Phamhi, Yamauchi, & Gómez-Sjöberg describes the introduction of a toolkit for assessing and harnessing crowdsourced annotations for imaging-based spatial transcriptomics datasets. Specifically, the authors provide a pipeline to process input images for efficient crowdsourced annotation and augment this with a tool to generate simulated in situ images that can aid in the annotation process. The authors introduce these tools with an analysis of parameters that influence success at the annotation task by the crowdsourced workers.

Overall, I really like this manuscript. Automated processing of spatial transcriptomic data is non-trivial and the nature of the computational task can make it challenging to rely on “out of the box” solutions for spot detection, especially between imaging platforms and/or in situ labeling technologies. Accordingly, deep learning approaches often are the most powerful options for this task. The toolkit introduced here is timely and packed with useful features for preparing imaging data for manual annotation—be it by crowdsourced workers or in-house experts—in order to generate “ground truth” datasets to train the learning models. As crowdsourced annotation is likely to become more and more common for this task, I expect INSTA to be a valuable resource to the growing imaging transcriptomics community.

With regards to the manuscript itself, I found it clear and well-written. The figures are clear and easy to follow. The GitHub pages that support INSTA, SpotImage, and the manuscript and well-organized, well-documented and contain useful demos and other tidbits of information to help potential users get going. The data analysis results are well-supported by the approaches taken. I support publication in PLoS Computational Biology, provided the following minor points are addressed:

Minor Points:

1. It would help to make it clearer in the main text how many images and spots are being considered when generating the summary statistics (precision, recall, etc.)

2. There is a statement in the manuscript ‘The figures and data for this project are available from https://github.com/czbiohub/instapaper.’ As written, I expected to find the input images used at this location, but that does not seem to be the case. I was able to find the input images in other of the other repos. It would be helpful if the authors could clarify this.

**Have all data underlying the figures and results presented in the manuscript been provided?**

Reviewer #1: Yes

Reviewer #2: Yes

PLOS authors have the option to publish the peer review history of their article (what does this mean?). If published, this will include your full peer review and any attached files.

Reviewer #1: No

Reviewer #2: No
---

## [Decision Letter · Decision Letter 1]

14 Jul 2021

Dear Dr. Gomez-Sjoberg,

We are pleased to inform you that your manuscript 'Validation and tuning of in situ transcriptomics image processing workflows with crowdsourced annotations' has been provisionally accepted for publication in PLOS Computational Biology.

Best regards,

Manja Marz

Software Editor

PLOS Computational Biology

Manja Marz

Software Editor

PLOS Computational Biology

Reviewer's Responses to Questions

**Comments to the Authors:**

Reviewer #1: All comments were addressed appropriately.

Reviewer #2: The authors have satisfactorily addressed my previous concerns. I support publication in PLoS Computational Biology.

**Have the authors made all data and (if applicable) computational code underlying the findings in their manuscript fully available?**

Reviewer #1: Yes

Reviewer #2: Yes

PLOS authors have the option to publish the peer review history of their article (what does this mean?). If published, this will include your full peer review and any attached files.

Reviewer #1: No

Reviewer #2: No

---

## [Editor Report · Acceptance letter]

4 Aug 2021

PCOMPBIOL-D-20-01316R1 

Validation and tuning of *in situ* transcriptomics image processing workflows with crowdsourced annotations

Dear Dr Gómez-Sjöberg,

I am pleased to inform you that your manuscript has been formally accepted for publication in PLOS Computational Biology. Your manuscript is now with our production department and you will be notified of the publication date in due course.

With kind regards,

Olena Szabo
